# Complex microparticle architectures from stimuli-responsive intrinsically disordered proteins

Stefan Roberts[1], Vincent Miao [1], Simone Costa[1], Joseph Simon[1], Garrett Kelly[1], Tejank Shah[2], Stefan Zauscher[2] & Ashutosh Chilkoti[1✉]

The controllable production of microparticles with complex geometries is useful for a variety of applications in materials science and bioengineering. The formation of intricate micro-architectures typically requires sophisticated fabrication techniques such as flow lithography or multiple-emulsion microfluidics. By harnessing the molecular interactions of a set of artificial intrinsically disordered proteins (IDPs), we have created complex microparticle geometries, including porous particles, core-shell and hollow shell structures, and a unique 'fruits-on-a-vine' arrangement, by exploiting the metastable region of the phase diagram of thermally responsive IDPs within microdroplets. Through multi-site unnatural amino acid (UAA) incorporation, these protein microparticles can also be photo-crosslinked and stably extracted to an all-aqueous environment. This work expands the functional utility of artificial IDPs as well as the available microarchitectures of this class of biocompatible IDPs, with potential applications in drug delivery and tissue engineering.

---

[1] Department of Biomedical Engineering, Duke University, Durham, NC 27708, USA. [2] Department of Mechanical Engineering and Materials Science, Duke University, Durham, NC 27708, USA. ✉email: chilkoti@duke.edu

The ability to control the formation of polymer micro-structures with complex architectures and spatially segregated regions is becoming increasingly important for applications in materials science and bioengineering[1,2]. Biocompatible microparticles have been used extensively in diagnostics[3–6], drug delivery[7–11], cell encapsulation and tissue engineering[12–21]. The functional properties of these microparticles are dictated by their size and shape, their internal microstructure, and the properties of their components. To expand the range of available applications, researchers have therefore focused on the fabrication of morphologically unique structures composed of functional materials. Though many complex microparticle geometries can be currently fabricated, the majority of them require sophisticated fabrication techniques such as multiple-emulsion microfluidics or flow lithography[5,12,22–25]. In addition, though a few research groups have used proteins as for microparticle fabrication[26,27], the vast majority of biocompatible microparticles are exclusively composed of synthetic polymers or biologically derived polysaccharides[1]. Hence, there remains a great need to not only develop new microarchitectures, but also to do so using alternative—biologically relevant—materials that expand the range of available material and functional properties.

Intrinsically disordered proteins (IDPs) are one such class of material[28–30]. Several sub-classes of IDPs containing low complexity sequences phase separate in aqueous solvents into an aqueous-immiscible phase—coacervate—in response to external stimuli such as temperature, ionic strength, or pH[30–33]. These proteins are of biological interest for their ability to form intracellular membraneless organelles within cells[34]—also called biomolecular condensates[29,35]—and are of technological interest for their ability to drive self-assembly into nano- and microscale compartments in aqueous solvents in vitro[36]. Elastin-like polypeptides (ELPs) are a class of artificial IDPs inspired by a natural IDP—tropoelastin[37–39]. Because ELPs consist of VPGXG repeats (Fig. 1a) found in tropoelastin, they are non-toxic and biocompatible, leading to their extensive application in drug delivery and tissue engineering[40–42]. When heated above their cloud point temperature ($T_{cp}$), they form ELP rich, insoluble coacervate droplets in an ELP poor aqueous phase (Fig. 1b, c). Their $T_{cp}$ can be precisely tuned by manipulating the amino acid sequence and chain length[31,43]. Their lower critical solution temperature (LCST) phase behavior is thermodynamically reversible with little to no thermal hysteresis; the coacervate phase formed above the $T_{cp}$ of an ELP reversibly dissolves upon cooling below the $T_{cp}$ (Fig. 1b). The interesting thermodynamic properties of these polymers have allowed ELP diblock and multiblock copolymers to be successfully self-assembled into an array of nanostructures and in at least one instance into a vesicular microparticle[40,44,45]. Using water-in-oil droplet microfluidics, we also recently demonstrated that the miscibility of different ELPs can be finely controlled to create spherical core-shell particles of ELP coacervates[46]. These coacervate droplets could be further crosslinked in situ by brief exposure to UV radiation by genetically encoding a UV-reactive non-canonical azidophenylalanine residue into the monoblock ELP[47].

Though the disordered regions of IDPs are considered the primary driving force for phase separation, we also recently established that periodically spaced structurally ordered domains within an IDP can alter the internal microarchitecture of the coacervate without loss of control over $T_{cp}$[48]. These partially ordered polymers (POPs) utilize ELPs as the structurally disordered backbone and incorporate oligoalanine helices—reminiscent of those within tropoelastin—that are periodically spaced within the ELP sequence (Fig. 1a). POPs exhibit LCST phase behavior similar to ELPs, but with two important differences: first, upon raising the solution temperature above the $T_{cp}$ where

the POP chains de-solvate and collapse, the helices in the POP create physical crosslinks between POP chains due to polymer domain swapping. Though POPs retain full thermal reversibility, their $T_{cp}$ upon cooling ($T_{cp}$-cooling) is significantly lower than the $T_{cp}$ upon heating ($T_{cp}$-heating) because of the extra thermal energy required to break the hydrophobic interactions between the oligoalanine domains. The thermal hysteresis of POPs creates a metastable region in the phase diagram (Fig. 1b). Second, instead of forming liquid-like coacervates with no internal microstructure such as that exhibited by an ELP (Fig. 1ci), POPs self-assemble into fractal-like microporous networks with adjustable stiffness and porosity (Fig. 1cii). Because ELPs and POPs exhibit highly controllable, thermally responsive phase separation and share sufficient sequence homology to be partially miscible, we show that mixtures of the two types of artificial IDPs can be used to create complex microarchitectures using only simple droplet microfluidics and stepwise heating and cooling.

## Results

**Bulk network architectures from ELP-POP mixtures.** The $T_{cp}$ of POPs and ELPs are tunable by the identity and mole fraction of the guest residue (X) in the VPGXG repeat unit, and the $T_{cp}$s of POPs are further tunable by the mole fraction of embedded oligoalanine helices. We therefore used different ratios of alanine (A) and valine (V) in the guest residue position in ELPs and POP and the helical content of POPs to span a range of $T_{cp}$s of ELPs and POPs from 20 °C to 50 °C (Supplementary Figs. 1–2 and Supplementary Table 1). This range of $T_{cp}$s allows us to use two distinct types of POP-ELP mixtures: (1) one in which the POP is designed to transition at a lower temperature than the ELP upon heating the mixture, and (2) another in which the POP is designed to transition at a higher temperature than the ELP. Within POP-ELP mixtures, the two IDP populations are not fully miscible, with distinct aggregation events observed for both populations.

To demonstrate the structural consequence of phase separation in system 1, we used a mixture of POP(V)-25% (where 'V' designates the guest residue amino acid and 25% designates the fraction of oligoalanine) with ELP($V_4A_1$). Above the $T_{cp}$-heating of POP(V)-25%, the IDP forms a stable, porous network as expected. Continued heating to above the $T_{cp}$ of the ELP causes ELP coacervates to form and grow until they are able to interact with the preformed POP network. Upon contact with the network, they become immobile, forming 'fruits' of ELP on a POP network 'vine' in solution as illustrated in the schematic (Fig. 1di) and confocal microscopy sections (Fig. 1ei–ii). While the size of the ELP globules is somewhat varied, the average volume of the globules is directly correlated with the concentration of ELP in solution without altering the POP network structure (Supplementary Fig. 3). The system can also be cooled below the ELP $T_{cp}$—but above the $T_{cp}$-cooling of the POP—and reheated with no change in the average globule size (Supplementary Fig. 3).

System 2, with ELP(V) designed to coacervate before POP ($V_1A_4$)-25%, forms an entirely different type of structure. Heating above the $T_{cp}$ of the ELP forms polymer coacervate droplets, as expected. Upon raising the temperature above the $T_{cp}$-heating of the POPs, however, the POPs do not form a microporous network but instead wet with the outer edges of ELP coacervate droplets and form a physically crosslinked interconnected porous shell, such that the entire structure consists of spherical ELP coacervate droplets encased in a lattice-like shell of the POP coacervate as illustrated (Fig. 1dii) and shown in confocal microscopy sections (Fig. 1eiii–iv). Due to the hysteretic nature of POPs and their significantly lower $T_{cp}$-cooling than $T_{cp}$-heating, subsequent

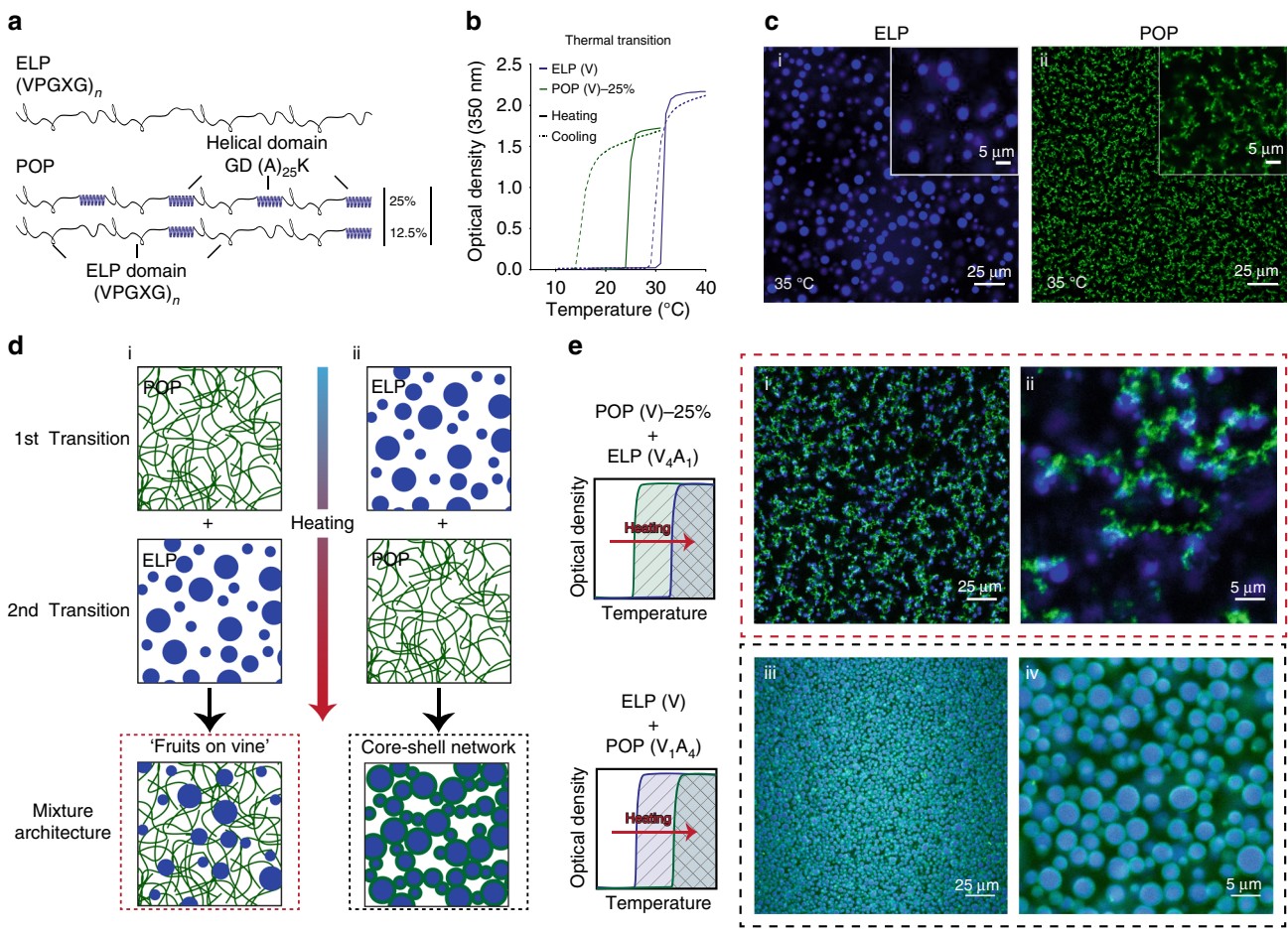

**Fig. 1 Unusual IDP architectures through ELP-POP mixtures. a** Sequence and architecture for ELPs and POPs. POPs have a disordered ELP backbone with ordered oligoalanine helices embedded at defined intervals. **b** Cloud point temperatures ($T_{cp}$) measured by optical turbidity for ELP(V) and POP(V)-25% (200 μM, PBS). Both IDPs have sharp, LCST phase behavior, though POPs exhibit thermal hysteresis with a lower $T_{cp}$-cooling than $T_{cp}$-heating. **c** Single plane confocal microscopy images above the $T_{cp}$s of (i) ELP(V) and (ii) POP(V)-25% (200 μM, PBS). While ELPs form liquid-like coacervates above their $T_{cp}$, POPs form stable, physically crosslinked porous networks. **d** Schematic of the formation of the (i) 'fruits-on-a-vine' and (ii) core-shell network architectures that form based on which component transitions at a lower temperature. **e** Single plane confocal images of (i-ii) mixtures of POP(V)-25% (200 μM) + ELP($V_4A_1$) (200 μM) depicting the 'fruits-on-a-vine' architecture and (iii-iv) mixtures of ELP(V) (500 μM) + POP($V_1A_4$)-25% (100 μM) depicting the core-shell architecture. Panels ii and iv are higher magnifications of images i and iii, respectively. Both mixtures were imaged after heating from 4 °C to 35 °C—above the $T_{cp}$ of both components—in PBS. Source data are provided as a Source Data file.

cooling of the system to $T < T_{cp}$ of the ELP but $T > T_{cp}$-cooling of the POP dissolves the ELP cores into the aqueous phase through the pores of the POP shell, and creates an interconnected network of hollow protein shells (Supplementary Fig. 4). These types of two-protein systems are a simple way to create a drug eluting scaffold using a single injectable system. To demonstrate proof-of-concept of this approach, we used fluorescence molecular tomography (FMT) to monitor the release of ELP($V_4A_1$) co-injected with POP(V)-25% in the subcutaneous flank of mice (Supplementary Fig. 5). The ELP "fruits" that hang from the POP "vine" slowly dissolve and are secreted out of the POP scaffold over 10 days without affecting the size of the POP scaffold. The release kinetics are further tunable with ELP concentration without altering the concentration of the co-injected POP.

**Production of atypical microparticle architectures**. Given the limited available architectures for biomaterial microparticles and the ease of formation for these atypical POP-ELP architectures in bulk, we next sought to translate these structures to microscale droplets. To do so, we used a microfluidic emulsion droplet generator in an X-junction design[46] (Fig. 2a) capable of producing

highly monodisperse water-in-oil emulsion droplets (Fig. 2b). The ELP and POP components in PBS were premixed and the entire device was kept at 4 °C during droplet generation to ensure uniform distribution of the soluble IDPs within each droplet. We controllably triggered subsequent ELP and POP phase transitions within the microdroplets by heating and cooling to generate a range of coacervate microstructures. We first examined the structures formed solely by POP(V)-25% in microdroplets, and found that the POP produces stable structures with micro-architectures similar to those observed in bulk (Fig. 2c–e and Supplementary Fig. 6), forming fractal-like porous microparticles with high void volume above the $T_{cp}$-heating. Continued heating and cooling above the aggregation temperature leads to nonlinear shrinking and swelling of the microparticles (Supplementary Fig. 7). Heated particles shrink/swell by as much as 20% between 20 and 50 °C, and the process is fully reversible. POP(V)-12.5% can also be used, forming identical microparticles with only a slightly higher $T_{cp}$-heating than the 25% POP (Supplementary Fig. 8). All particles remain stable when cooled into the meta-stable hysteretic temperature range, and subsequent cooling below the $T_{cp}$-cooling of the POP results in complete dissolution of the microstructure.

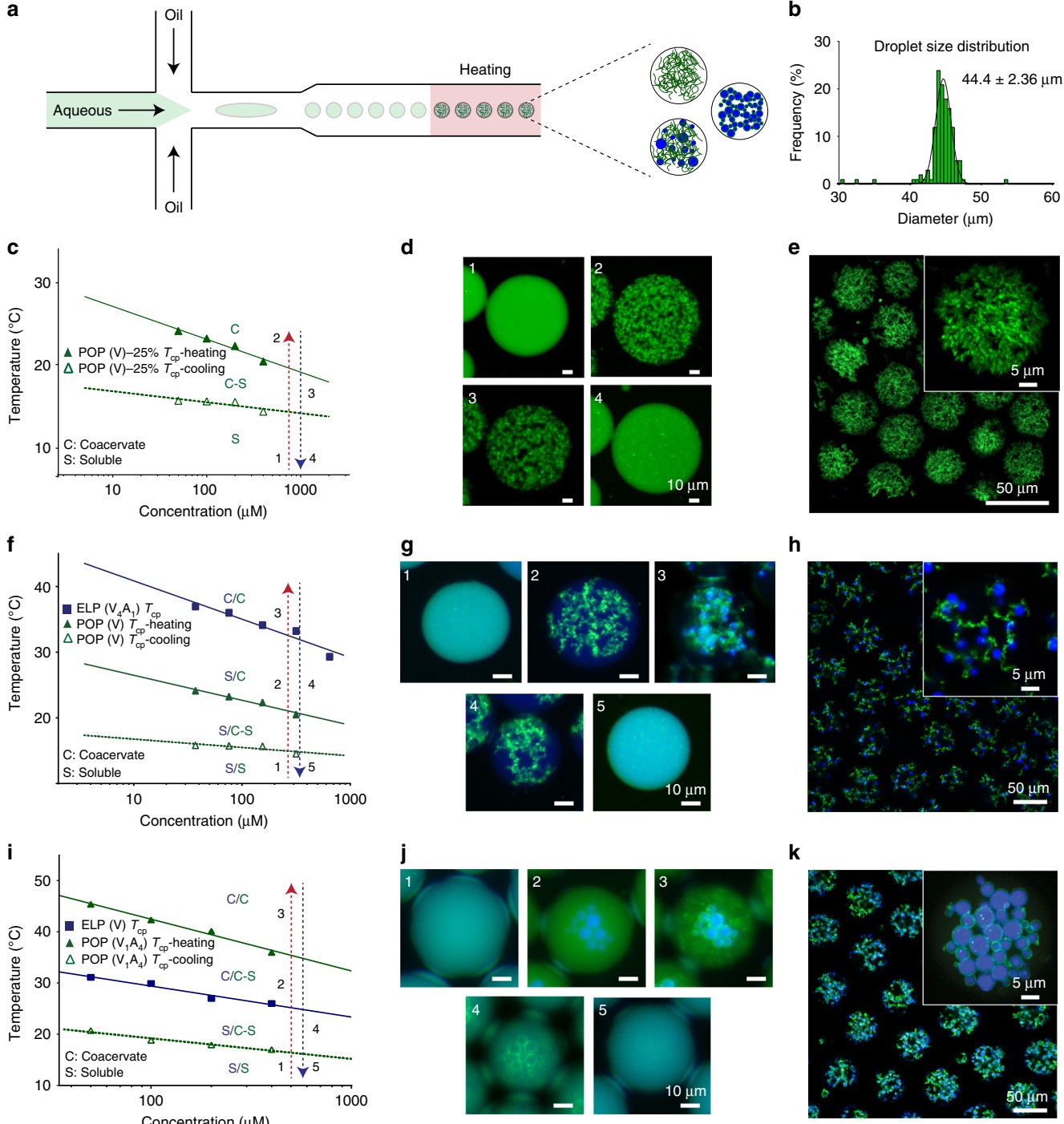

**Fig. 2 Microdroplet architectures. a** Depiction of the microfluidic device used to generate microparticles. **b** Image analysis of droplets ($n = 125$ droplets) reveal a high degree of monodispersity. **c** Partial phase diagram for POP(V)-25% illustrating the different discrete states possible during a heating and cooling cycle. **d** Fluorescence images of POP(V)-25% (500 μM) microdroplets during a heat-cool cycle through the states shown in (**c**). The metastable hysteretic range prevents dissolution of the particles until the solution temperature is lowered below the $T_{cp}$-cooling. **e** Confocal images (25 μm stack) of the same particles in the metastable hysteretic state—state 3. **f** Partial phase diagram for mixtures of POP(V)-25% (200 μM) + ELP($V_4A_1$) (200 μM) depicting the different discrete states possible during a heat-cool cycle of this system in which the POP aggregates at a temperature below the ELP. **g** Fluorescent images of each state of the cycle demonstrating the formation of the fruits-on-a-vine architecture in state 3. **h** Confocal images of state 3 clearly depicting the ELP "fruits". **i** Partial phase diagram of ELP(V) (500 μM) + POP($V_1A_4$)-25% (100 μM) depicting the states available during a heat-cool cycle. **j** Fluorescent images of each state of the cycle. Due to the hysteretic nature of the POP and its lower $T_{cp}$-cooling than the $T_{cp}$ of the ELP, the ELP dissolves first upon cooling, diffusing out and leaving a network of hollow POP shells. **k** Confocal images of core-shell networks formed in state 3. Source data are provided as a Source Data file.

We next included ELP as a second polymer component within the aqueous phase of our microfluidic set-up to recapitulate the unique architectures seen in bulk within a confined microenvironment with the end goal of creating microparticles with unique morphologies and internal microstructures. Mixtures of POP(V)-25% and ELP($V_4A_1$), in which the POP phase separates at a lower temperature than the ELP, were thermally programmed into the five distinct states seen in the overlaid ELP and POP phase diagram (Fig. 2f–h and Supplementary Fig. 6): (1) ELP and POP are soluble; (2) upon heating to a temperature > $T_{cp}$-heating of the POP, the POP phase separates and forms a porous microparticle; (3) upon continued heating to $T > T_{cp}$ of the ELP, the ELP coacervates into immiscible globules that wet the POP network; (4) upon cooling below the $T_{cp}$ of the ELP, the ELP dissolves; (5) and further cooling to $T < T_{cp}$-cooling of the POP cooling also re-dissolves the POP.

In contrast, mixtures of ELP(V) + POP($V_1A_4$)-25% in which the ELP coacervates at a lower temperature than the POP ($T_{cp}$ ELP < $T_{cp}$-heating POP), can be cycled through core-shell and hollow shell networks (Fig. 2i–k and Supplementary Fig. 6) as follows: (1) both IDPs are fully dissolved; (2) as the temperature is raised such that $T > T_{cp}$ of the ELP, the ELP coacervates into aqueous-immiscible droplets; (3) raising the temperature to $T > T_{cp}$-heating of the POP triggers the phase separation of the POP, leading to the formation of a conformal porous POP shell on the ELP core; (4) upon cooling to $T < T_{cp}$ of the ELP but $T > T_{cp}$-cooling of the POP, the ELP dissolves resulting in a network of hollow POP shells; (5) finally, as the temperature is lowered below the $T_{cp}$-cooling of the POP, the POP re-dissolves, fully restoring the system to its original state of a mixture of soluble ELP and POP. Like the porous POP microparticle networks, the hollow POP shells also swell and shrink ~20% in size when heated and cooled after formation (Supplementary Fig. 7). After reaching state 4, where the ELP has dissolved out from within the POP shells into the aqueous phase of the droplets, leaving behind intact porous POP shells, if the temperature is then raised above the $T_{cp}$ of the ELP, the ELPs will re-coacervate, forming aqueous-immiscible ELP globules that wet the outside of the hollow POP shells (Supplementary Fig. 9).

**Incorporation of unnatural amino acids (UAAs) for UV crosslinking**. To augment their stability, we next devised a method to crosslink the microstructures without the need of extrinsic crosslinking agents, and without the formation of potentially toxic byproducts. To do so, we pursued multi-site UAA incorporation of *para*-azidophenylalanine (*p*AzF), which participates in a host of crosslinking reactions following exposure to ultraviolet (UV) light[49]. Following the recent successes in engineering *E. coli* strains optimized to site-specifically express UAAs with high fidelity and yield, as well as their use in fabricating thermally responsive micro-gels[47,50], a small library of UV crosslinkable xPOPs was created and their thermal and microarchitecture properties were characterized (Supplementary Fig. 10). xPOPs are identical to POPs in their sequence, with the exception that *p*AzF residues are equally spaced throughout the polymer at 1 *p*AzF per 100 residues. While the $T_{cp}$-heating and $T_{cp}$-cooling of the xPOPs are slightly depressed relative to the parent POP due to the hydrophobicity of *p*AzF, the xPOP microemulsions undergo an identical coacervation process, forming porous microparticles. The xPOPs photochemically react after only short UV exposure, requiring <10 s of exposure time to fully crosslink. Network architecture and void volume in bulk are unaffected by crosslinking (Supplementary Fig. 10). When crosslinked above their $T_{cp}$-heating, subsequent cooling to below

$T_{cp}$-cooling does not resolubilize the microparticles, unlike their non-crosslinked counterparts (Supplementary Fig. 11).

xPOPs can also be readily mixed with ELPs to stabilize microparticle architectures. Mixtures of xPOP(V)-25% and ELP ($V_4A_1$) form a fruits-on-a-vine architecture when heated above the IDPs' aggregation temperatures similar to that previously observed for a POP and ELP mixture (Supplementary Fig. 12). Subsequent exposure to UV light crosslinks the POP, allowing the ELP to solubilize, but preventing the POP from solubilizing even when cooled well below its $T_{cp}$-cooling. If the system is re-heated, ELP globules reform, and this process can be repeatedly cycled without altering the POP microparticle architecture. Cooling and re-heating does not alter ELP globule properties, and the average "fruit" size remains unchanged. Compared to bulk mixtures, the ELP "fruits" formed in microparticles are similar in shape but slightly larger in size ($p < 0.05$, Student's *t*-test, $n = 50$).

**Using heating rate to control microarchitectures**. Within the polymer sequence framework presented in this manuscript, the order in which the two components—ELP and POP—phase separate controls the type of architecture that is formed, rather than the specific sequences of the chosen POP and ELP. For example, mixtures of ELP(V) + POP($V_1A_4$)-12.5% (Supplementary Fig. 13) and ELP(V) + POP($V_1A_1$)-25% (Supplementary Fig. 9) both form core-shell structures similar to ELP(V) + POP ($V_1A_4$)-25% (Fig. 3) despite the differences in all three POP sequences. However, these structures are not wholly identical. We determined that the smaller the gap in transition temperatures between the ELP and POP, the smaller the resulting individual core-shell structures that make up the network. This observation highlights a critical difference between the combination of non-hysteretic ELP and hysteretic POP and our previous work on dual emulsion ELPs[46,47]. Given sufficient time, two immiscible ELPs with different transition temperatures will phase separate from one another into identical structures regardless of when the second ELP transition is triggered. In the ELP-POP system, increasing the temperature range of thermal hysteresis—the $T_{cp}$ gap—also increases the amount of time that ELP is given to coalesce at a constant thermal ramp rate, prior to entrapment by POPs. Given the sequence homology between the disordered components of POPs and ELPs, POPs prefer to aggregate around the ELPs, and once even a very small layer of POP has formed around the ELP, the ELP coacervate droplets can no longer continue to coalesce.

These results suggested that the core-shell architecture can be controlled by heating ELP and POP mixtures at different rates. To investigate this, we used mixtures of ELP(V) and xPOP($V_1A_4$)-12.5% to illustrate the spectrum of core-shell structures achievable (Fig. 3a–c and Supplementary Fig. 15). ELP + xPOP mixtures were heated at ramp rates of 0.5–20 °C/min in a thermocycler from 4 to 50 °C and were then UV-crosslinked at the final temperature. Samples were then cooled and transferred to a fluorescent microscope for imaging. At high ramp rates, the ELP is given limited time to coalesce, resulting in disperse ELP globules that become encapsulated by conformal porous shells of the POP, producing a large network of small hollow crosslinked shells. Slightly faster ramp rates produce similar networks, with slightly larger POP shells and some "network-like" arms likely due to (a) the absence of ELP and (b) insufficient time to interact with already aggregated POP. At 1 °C/min, a bimodal distribution emerges with an average of one large and one small shell per aqueous droplet. Notably, when the heating rate is slowed to 0.5 °C/min, a single hollow-spherical POP shell per droplet is formed. Reheating does not re-fill the shells (Supplementary

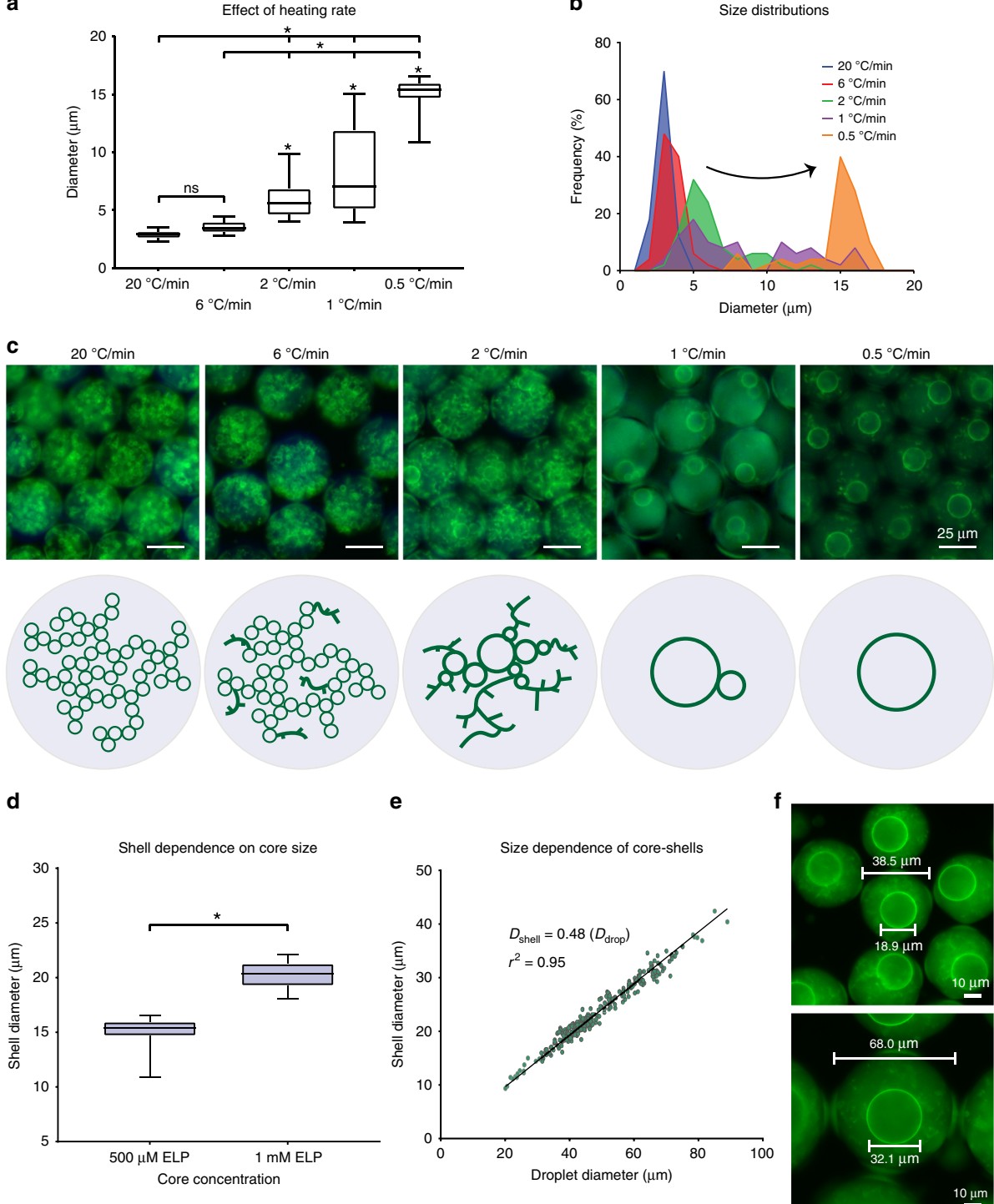

**Fig. 3 Controlling hollow shell architecture. a** Size distribution of xPOP shells formed after heating a mixture of ELP(V) (1 mM) + xPOP(V$_1$A$_4$)-12.5% (100 μM) at different constant rates (10–90% box and whiskers plot with median central line bounded by 25 and 75% quartiles, *$p < 0.05$ as determined by one-way ANOVA with Tukey's post hoc test, $n = 50$ shells measured using 3D confocal image stacks for each rate). **b** Histogram of the size distribution of the xPOP shells shows a broad distribution of shell diameters at ramp rates > 1 °C, the development of a bimodal distribution at 1 °C/min, and emergence of a unimodal size at 0.5 °C/min. **c** Fluorescence microscopy images and corresponding cartoon of the hollow xPOP shell architectures that form at different heating rates. The system shifts from a network of interconnected hollow shells to a single hollow protein shell as the heating rate is slowed. **d** Hollow xPOP protein shell diameter increases with increased ELP concentration (10–90% box and whiskers plot with median central line bounded by 25 and 75% quartiles, *$p < 0.01$ as determined by two-tailed Students $t$-test, $n = 50$ droplets for each rate). **e** Linear regression analysis for a polydisperse mixture of ELP(V) (1 mM) + xPOP(V$_1$A$_4$)-12.5% (100 μM) comparing diameter of the water-in-oil emulsion droplets with the diameter of the xPOP shells contained within the droplets ($n = 280$ droplets). **f** Typical fluorescence images illustrating the linear correlation between droplet diameter and xPOP shell diameter and the ~0.5 scaling pre-factor that relates droplet diameter to xPOP shell diameter. Source data for are provided as a Source Data file.

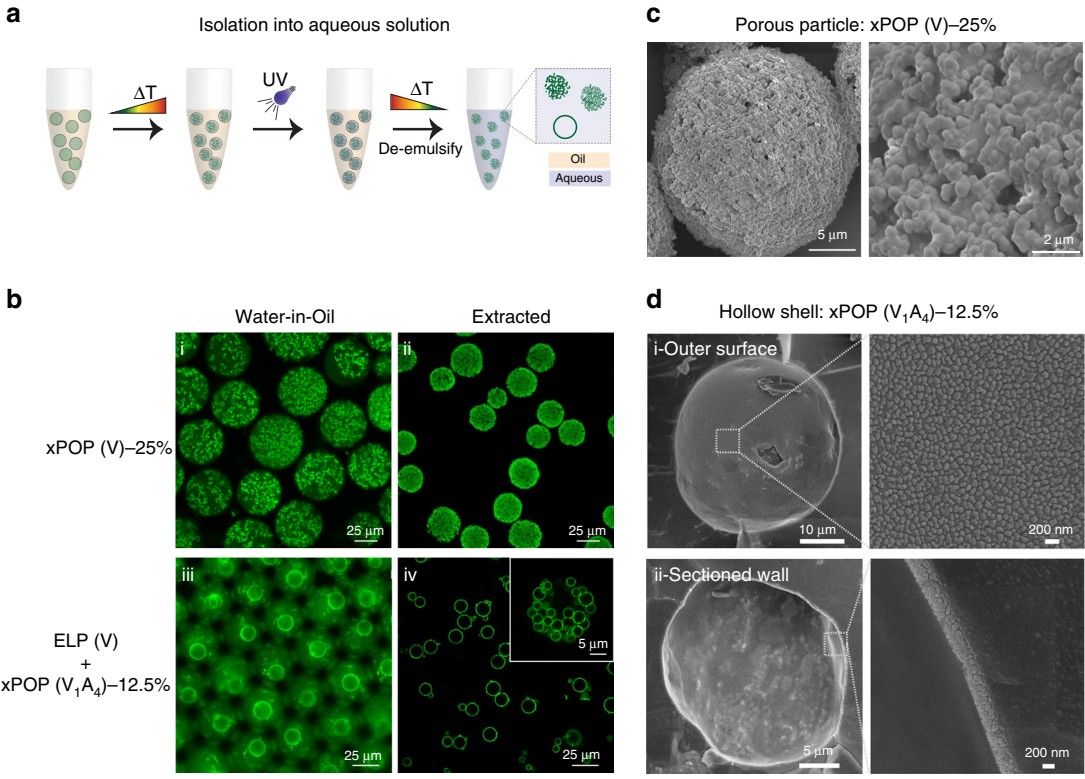

**Fig. 4 Extraction into an aqueous environment. a** Schematic of the process of extraction from water-in-oil to an all aqueous (buffered saline) environment. **b** Microscopy images of (i) unextracted (500 μM) and (ii) extracted xPOP(V)-25% porous microparticles and (iii) unextracted and (iv) extracted xPOP shells formed from mixtures of ELP(V) (1 mM) + xPOP (V₁A₄)-12.5% (100 μM) mixtures. The inset in iv demonstrates that hollow shell networks created with faster heating rates can also be extracted. **c** SEM of a xPOP(V)-25% microparticle showing the interconnected coacervate architecture that comprises the networked particle. **d** Cryo-SEM images of (i) the outer surface and (ii) a fractured xPOP (V₁A₄)-12.5% shell. The walls of each shell range from ~200–400 nm thick are composed of tightly packed nano-coacervates of xPOP.

Fig. 14), but it does cause aggregation of ELP within and outside of the hollow spheres.

Not only can the architecture of these POP shells be controlled, from a network or hollow protein shells to a single hollow protein shell, but we can also control their size (Fig. 3d–f and Supplementary Fig. 15). If the diameter of the aqueous droplet is kept constant, doubling the volume of ELP within each aqueous droplet, for example, also doubles the volume within a POP shell. The diameter of the droplet provides an even more convenient way of controlling size. By varying the speed of the aqueous phase during droplet generation, and therefore creating a polydisperse mixture of droplet sizes, we were able to linearly correlate the diameter of the resultant POP shell to the droplet diameter. With 1 mM ELP(V) as the core-forming component, the diameters of the POP shells were consistently half that of the particle diameter (Fig. 3e, f).

**Microparticle extraction into an all-aqueous environment.** For many downstream applications, POP microparticles must be extracted back into an aqueous environment from the emulsion. In addition to stabilizing the microstructures that are formed within water-in-oil emulsions, UV crosslinking through UAA incorporation allows extraction of these microstructures into an all aqueous phase (Fig. 4a). Using a simple de-emulsification process (see Methods), porous POP microparticles were successfully recovered into buffered saline (Fig. 4b, c). The particles are mechanically stable enough to retain their shape and porosity, albeit with a 40% reduction in their size following extraction

(40.7 μm ± 1.8 μm pre-extraction, and 25.3 μm ± 3.4 μm post extraction, $n = 50$ particles each).

To determine their mechanical stability, we used high frequency atomic force microscopy (AFM) to evaluate the Young's modulus (E) (Supplementary Fig. 16) of extracted POP microparticles and controls—planar bulk gels of crosslinked ELP (xELP(V)) and a crosslinked POP with 25% helical content (xPOP(V)-25%). While the planar xELP(V) has an E of $0.12 \pm 0.02$ kPa, the additional stabilization conferred by physical crosslinking increases E of the bulk xPOP(V)-25% by almost one order of magnitude to $1.4 \pm 0.3$ kPa. Notably, the micro-particles of the xPOP(V)-25% were more than an order of magnitude stiffer than the bulk material, with xPOP(V)-25% particles reaching a Young's modulus of $20.9 \pm 2.8$ kPa, indicating that the ~40% compaction in the size of the microparticles that occurs post-extraction has an unexpectedly significant effect on its mechanical stiffness. However, the degree of helix incorporation did not have a statistically significant effect on the Young's modulus of crosslinked microparticles, as the Young's moduli of xPOP(V)-25% and xPOP(V)-12.5% were not statistically different, suggesting that any additional physical stabilization conferred by the increased helical content is overwhelmed by the effect of chemical crosslinking and their physical compaction upon extraction from the water-in-oil emulsion.

Crosslinked hollow shells and hollow shell networks can be extracted using the same method as for POP only microparticles (Fig. 4b, d). Despite their thin walls, they are sufficiently strong to tolerate the extraction procedure, though they require hydration to maintain their spherical shape. Imaging the hollow POP shells

by cryo-SEM reveals their morphologically rich multiscale architecture. Their remarkably thin walls (Fig. 4dii), which range in thickness from 200–400 nm, are composed of tightly packed nanoscale coacervates of POP. These coacervates are architecturally similar to those that make up the bulk POP networks and microparticles. Perhaps due to the influence of the templating ELP core-component, however, they coarsen on a more rapid time scale than the POP microporous networks, allowing tighter formation of smaller coacervtes into a thin, interconnected layer. This packing—which looks similar to tire threads at the outer surface (Fig. 4di)—is the source of the shells' porosity. ELP coacervates that originally formed the core of the shells are, once cooled below their $T_{cp}$ and solubilized, able to traverse the interconnected pathways between the interconnected POP coacervates and diffuse into the surrounding aqueous phase.

## Discussion

Using a set of de novo recombinant IDPs that were designed to exhibit with specific LCST phase behavior, we have created unique microarchitectures within microparticles by combining simple, scalable processing techniques with temperature-responsive phase behavior. All of these architectures are unique in protein-based microparticles and some, such as the 'fruits-on-a-vine' and hollow microshell networks, are, we believe, unique across all classes of biocompatible materials. These microparticles can be further chemically crosslinked by the incorporation of a genetically encoded UV crosslinkable UAA and brief UV-irradiation of the droplets. After crosslinking, the covalently stabilized microparticles can be extracted from the water-in-oil emulsion into an all aqueous environment, to provide stable, non-aggregating microparticles. In conclusion, this work demonstrates that rationally designed artificial IDPs with tunable and programmed aqueous demixing phase behavior can be readily combined with conventional microfluidic polymer processing technology to create materials with potentially diverse applications.

## Methods

**Gene synthesis**. ELPs and POPs: Single stranded oligonucleotides encoding the polymer genes were purchased from Integrated DNA Technologies (IDT) and cloned into a modified pet-24 vector via recursive directional ligation by plasmid reconstruction, as previously described, into chemically competent Eb5α E. coli to assemble the full-length polymer genes[7]. In brief, A and B populations of each gene fragment were generated by restriction digest with AcuI and BglI and BseRI and BglI, respectively. Ligation of appropriate plasmid fragments from A and B populations following DNA gel purification resulted in the formation of a single, concatenated A + B gene fragment inside the modified pet-24 vector.

xPOPs: Following their full-length assembly using the above methods, xPOP genes were further isolated via BseRI and BamHI restriction digest, and the isolated gene was cloned into another modified vector with a pTac promoter and rrnB terminator instead of the T7 promoter and terminator of the original vector. The plasmids were then co-transformed into c321.ΔA E. coli alongside a pEvol tRNA/aaRS vector with two copies of pAcFRS.1.t1 synthetase. The C321.ΔA genome has previously been edited to remove all instances of the amber stop codon, and the tRNA/aaRS pair has been optimized to recognize the amber stop codon and incorporate para-azidophenylalanine[47,50].

**Biopolymer synthesis and characterization**. ELP expression: Liquid cell cultures from 25% glycerol stocks were grown overnight (~16 h) in 25 mL 2xYT starter cultures containing 45 µg/mL kanamycin at 37 °C and 200 rpm. Starter cultures were then transferred to 1 L 2xYT cultures the following morning and grown for ~8 h at 37 °C at 200 rpm in the presence of 45 µg/mL kanamycin. One millimolar IPTG was then added to induce expression, and cultures were grown for an additional ~16 h overnight at 37 °C and 200 rpm.

POP expression: POP expression follows a nearly identical protocol as ELPs, however, cultures are grown at 25 °C prior to induction with IPTG and at 16 °C overnight after induction to reduce the formation of truncation products from ribosomal pausing[48].

xPOP expression: xPOP expression follows a similar protocol with two exceptions. (1) 45 µg/mL kanamycin, 25 µg/mL chloramphenicol, 0.2% arabinose, and 1 mM pAzF are included in all cultures from inoculation, (2) cultures are grown at 34 °C, determined to be the optimal temperature for pAzF incorporation[50].

Purification: For all IDPs, cell pellets were collected via centrifugation at 3500 × g for 10 min and resuspended in PBS to an appropriate volume (~25 mL). Samples were sonicated for a total of 3 min to lyse cells, and supernatant was collected following centrifugation at 23k × g for 10 min at 4 °C. 2 mL/L culture of 10% PEI was added, and the supernatant was collected following centrifugation at 23k × g for 10 min at 4 °C. Three rounds of inverse transition cycling (ITC), a method that utilizes the thermally responsive properties of the IDPs[51], were then performed to purify the IDPs. Briefly, samples were heated to 50 °C (supplemented with 2 M NaCl in the first cycle) and centrifuged at 23k × g and 30 °C for 10 min. Then, pellets were resuspended in PBS and put through another centrifugation step at 23k × g and 4 °C for 10 min in each cycle. Purity was determined via SDS-PAGE gel electrophoresis. Samples were then dialyzed into water, lyophilized, and stored at −20 °C. All protocols for xPOP expression and purification were completed under low-light conditions to avoid undesirable pAzF crosslinking during synthesis and purification.

**Turbidity**. Coacervation behavior was characterized with WinUV using a Cary 100 UV-Vis spectrophotometer monitoring optical density at 350 nm (or 650 nm for xPOPs as pAzF absorbance interferes at 350). Samples in 1x PBS were heated and cooled at 1 °C/min and the temperatures at which the first derivative of the curve was the maximum were defined as the $T_{cp}$ -heating and -cooling.

**Microparticle synthesis and extraction**. Microparticles were generated in water-in-oil emulsions using a quartz X-junction droplet microfluidics device (Dolomite) similar to our previous setup[46]. In brief, precision syringe pumps were used to flow an aqueous phase at 50–150 µL/h (variation in flow rate was used to control size of the microparticles) and an oil phase composed of 75%/5%/20% v/v TEGOSOFT DEC/ABIL EM 90/mineral oil at 250 µL/h through a polydimethylsiloxane (PDMS) microparticle generator. Microparticles were generated at 4 °C to prevent polymer aggregation during the process. Microemulsions were stable for ~1 week when stored at 4 °C before loss of monodispersity. Where necessary, an Omincure Series 1000 lamp was used to crosslink samples (30 s, 50% power, 100 W lamp, 312 nm) in emulsion.

To extract microparticles back into an aqueous environment, water-in-oil emulsions were gently mixed 1:10 v/v into isobutanol, and hand-rotated for 30 s. Samples were centrifuged at 80 × g at 4 °C for 5 min to pellet microparticles, and the oil phase removed. Microparticles were then twice washed with 1× PBS and centrifuged at 80 × g for 5 min. Microparticles were then resuspended in 1× PBS for microscopy.

**Microscopy**. The IDPs were fluorescently labeled using either Alexa Fluor 350 or Alexa Fluor 488 NHS Ester with a typical reaction efficiency of 50%. Excess dye was removed with dialysis and IDPs were lyophilized for storage. For all experiments, the fluorescently labeled IDPs were mixed with the unlabeled IDP such that <10% mole fraction of POPs in solution were labeled.

Optical fluorescence microscopy was performed on an upright Zeiss Axio Imager A2 microscope with a Zeiss Incubation System S heating stage and PeCon TempController 2000-2 temperature control unit. Unless otherwise stated, samples were heated and cooled at 5 °C/min. Sample drying was problematic at higher temperature with heating rates of 0.5 °C/min and 1 °C/min. As a result, these samples were heated and cooled in an Arktik thermal cycler and then rapidly (<10 s) transferred to the microscope stage that was pre-heated to the final desired temperature. Confocal microscopy was performed on a Zeiss 710 inverted confocal microscope with an environmental heating chamber. All samples were imaged at either 25 °C or 35 °C after equilibration at the appropriate temperature. All images were captured with Zen (Zeiss) and processed and analyzed in ImageJ. Where possible, a standardized image processing cascade using thresholding and the 'analyze particles' plugin was used to automate size calculations. In rare cases, manual line segments were also drawn.

Scanning electron microscopy was performed on a FEI XL30 scanning electron microscope using Scandium (ResAlta) for image acquisition. Microdroplets extracted into PBS were allowed to dry at room temperature and sputter coated with gold prior to imaging. Cryo-SEM was performed on a JEOL JSM-7600F SEM outfitted with a cryogenic transfer system. Samples were flash frozen in liquid nitrogen slush and transferred under vacuum to the preparation chamber where the sample was fractured and etched under vacuum. Samples were then sputter coated with gold for imaging. All images were processed and analyzed in ImageJ.

**Fluorescence molecular tomography**. ELP($V_4A_1$) was fluorescently labeled with Alexa Fluor 647 NHS ester with a reaction efficiency of ~50%. Excess dye was removed by ultrafiltration using Amicon Ultra Centrifugation Filters. Labeled ELP was mixed with unlabeled ELP to obtain a final fluorophore concentration of 1 µM. Prior to injection, POP(V)-25% and ELP($V_4A_1$) were endotoxin purified to < 1 EU/ml and mixed to a final POP concentration of 250 µM and ELP concentrations of 10 µM, 100 µM, and 250 µM. C57BL/6J mice at 8 weeks old were shaved below the midline and injected subcutaneously on the right hind flank with 200 µL of the POP-ELP mixture corresponding to the appropriate group. At 0 (immediately after injection), 4, 12, 24, 48, 72, 108, 144, 192, and 240 h post-injection, mice were anesthetized with 2.5% isoflurane and imaged with a Fluorescence Molecular Tomography 4000 In Vivo Imaging System. Quantification of fluorescence in the region of interest was performed using TruQuant software.

The Duke University Institutional Animal Care and Use Committee (IACUC) approved and provided oversight for the use of animals within the manuscript. All animals were housed in a dedicated rodent facility kept at ~20 °C and ~50% humidity. Animals were provided food and water ad libitum and 12 h/12 h light/dark cycles by facility staff.

**Atomic force microscopy (AFM).** A commercial Asylum MFP-3D system and Asylum Research were used for all experiments. All experiments were conducted in PBS at room temperature. Drift was minimized by equilibrating the system at least 15 to 30 min in solution prior to any measurement. Five micrometer borosilicate beads were attached to AFM cantilevers with a spring constant 0.58 N/m. An optical microscope and a micromanipulator were used to apply UV cure epoxy and borosilicate beads to tipless cantilevers. After the beads were adhered to the cantilever, they were cured by UV irradiation at 366 nm for 90 min. Excess epoxy was removed by reactive ion etching and serial rinsing in a 1% (v/v) sodium dodecyl sulfate solution in deionized water solution, deionized water, and ethanol. Only attached beads with defect free surfaces, as confirmed by scanning electron microscopy, were used for subsequent AFM measurements. Cantilevers modified with attached beads were functionalized with a 2 nm Cr layer and a 10 nm Au overlayer by E-Beam evaporation. Coated probes were then incubated overnight with triethylene glycol to form a uniform nonfouling monolayer[52]. The deflection sensitivity was calibrated by engaging the cantilever on a silicon surface in deionized water. The spring constant, $k_c$, of the cantilever was determined from the power spectral density of the thermal noise fluctuations in air by fitting the first free resonant peak to known equations for a simple harmonic oscillator[53]. All data processing and calculations were performed in MATLAB. The contact point between the probe and sample in force curves were identified visually and used to offset the force curves to 0 at the contact point. A Hertzian contact mechanics model was fitted to these curves to calculate the Young's modulus (E)[54]. For each individual microparticle, an estimate of particle radius was obtained by ImageJ analysis of images collected from under the AFM video feed. For planar surfaces, a radius of infinity was used in the model. All samples were adhered to glutaraldehyde activated glass coverslips in PBS and checked under the microscope with gentle agitation to ensure rigid coupling to the underlying substrate.

**Statistics and reproducibility.** All statistical analysis was carried out using GraphPad Prism 8. When comparing individual groups, two-tailed $t$-tests were used to determine statistical significance. ANOVA was used to evaluate significance among three or more groups and with appropriate post hoc tests where indicated in the text for comparisons between groups. For animal experiments, the resource equation method was used to determine the minimum number of mice for each group[55]. For particle analysis and measurements, the largest feasible group size captured within at least three independent imaging windows was chosen with a minimum experimental number of 50 measurements. Exceptions to this rule were the bulk network void volume measurements and particle video analysis, where an $n = 3–5$ was chosen. Specific experimental group sizes are reported in the description of each experiment. All turbidity measurements (Figs. 1b, 2c/f/i, Supplementary Figs. 1, 6a–c, 10b, 12a) were repeated at least three times with similar results. Fluorescence, confocal, and SEM microscopy imaging and analysis (Fig. 1c/e, 2b/d/e/g/h/j/k, 3, 4b-d, Supplementary Figs. 3, 4, 6, 7, 8, 9, 10 f/g, 11, 12b–e, 13, 14, 15) were repeated at least three times with similar results, and all microparticle images are representative of their broader population. AFM (Supplementary Fig. 16) was repeated twice, and in vivo experiments (Supplementary Fig. 5) were only performed once. Polymers were purified several times from independent stocks to ensure observed behavior was not batch dependent. No differences in polymer batches were observed.

**Reporting summary.** Further information on research design is available in the Nature Research Reporting Summary linked to this article.

## Data availability
The authors declare that all data supporting the findings of this study are available within the manuscript and its supplementary files and are available from the authors on reasonable request. Source data underlying Figs. 1b, 2b/c/f/i, 3a/b/d/e and Supplementary Figs. 1b/c, 2, 3a/d, 5a, 7a/b/d, 10b/d/f, 12a/d/e, 15d, 16c/d are further provided as a Source Data file.

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

## Acknowledgements

This work was funded by the NIH through MIRA grant # R35GM127042 to A.C., by the NSF through the Research Triangle MRSEC (NSF DMR-11-21107), and by the NSF Graduate Research Fellowship Program under Grant No. 1106401. This work was performed in part at the Analytical Instrumentation Facility (AIF) at North Carolina State University, which is supported by the State of North Carolina and the National Science Foundation (award number ECCS-1542015). The AIF is a member of the North Carolina Research Triangle Nanotechnology Network (RTNN), a site in the National Nano-technology Coordinated Infrastructure (NNCI).

## Author contributions

S.R. designed and performed experiments, analyzed data, and prepared the manuscript. V.M. constructed the xPOP library and designed and performed experiments. S.C. optimized UAA incorporation and designed experiments related to its use. J.S. optimized the microfluidics design and aided in microparticle design and production. G.K. helped design and carry out in vivo experiments. T.S. designed and carried out the AFM experiments. S.Z. provided guidance on the mechanical analysis of microparticles. A.C. provided guidance, designed experiments, and prepared the manuscript. All authors participated in data discussion and manuscript preparation.

## Competing interests

The authors declare no competing interests.
