## [Peer Review File · Nature Communications]

Reviewers' comments:

Reviewer #1 (Remarks to the Author):

This manuscript describes the fabrication of protein microparticles with complex internal structures by basic microfluidic processing techniques. Two polypeptides that exhibit temperature induced phase transitions are used and the differences in the phase transition temperatures of different protein variants, as well as protein concentration and heating rate, are used to produce several different microstructures. The ability to produce several different protein microstructures with the same class of molecules is unique, though the authors do not acknowledge prior reports of similarly structured protein microparticles made from different proteins and simple processes. Without demonstration of the utility of any of the reported structures, it is difficult to gauge the impact of these particular materials relative to more simple protein or polymer microparticles with no internal structure that are common in the literature, or the few reports of complex-structured protein microparticles. It may be possible to extend this phase transition methodology to other types of polymers and proteins that have temperature (or other) triggered transitions, thereby providing a more general framework for complex microstructure formation. Specific comments are listed below.

Major:

1. While the geometries are beautiful, it is not demonstrated or described what value these protein microparticles have. It is mentioned that there are "potential applications in drug delivery and tissue engineering" but there is no discussion of why these complex morphologies would be needed relative to simple spherical protein or biopolymer/carbohydrate hydrogel microparticles that are so prevalent in the literature, or practical demonstration of the utility of complex microstructure. Presumably, to be used in bio-applications the microparticles will need to be loaded with functional molecules or cells in a way that is useful relative to the local micro-geometry. Will these additives affect the ability to form the structures?
2. The first half of the following sentence (in the "Discussion"), from which the authors derive novelty, is not true. "All of these architectures are novel in protein- or polymer-based microparticles ..." The manuscript did not cite previous work making protein microparticles with complex morphologies under simple conditions with morphologies similar to those described here. A literature search finds relevant three papers from the last 7 years, including microclusters of protein spheres (Song. "Budding-like division of all-aqueous emulsion droplets modulated by networks of protein nanofibrils." *Nature communications* 9.1 (2018): 2110.), hollow shell and core shell protein microstructures (Park. "Thermally triggered self-assembly of folded proteins into vesicles." *Journal of the American Chemical Society* 136.52 (2014): 17906-17909.), and porous protein microparticles (Volodkin. "One-Step Formulation of Protein Microparticles with Tailored Properties: Hard Templating at Soft Conditions." *Advanced Functional Materials* 22.9 (2012): 1914-1922.) While the literature in this area is sparse and describes single or dual morphologies, and this manuscript is capable of making multiple morphologies with the same general system, previous work should be acknowledged and used to place the current work into context. Indeed, as noted in this manuscript, in the polymer field published complex microstructures seem to be the result of complex processes.
3. It is very difficult to tell that the images and cartoons in Fig 3c match. The POP structures appear very fuzzy. There is no visually detectable difference between 20 and 6 degrees C/min heated samples. The smaller shell drawn in the cartoon and reported in Fig 3a,b is only visible in one particle for 1 degree C/min heating. The 0.5 degree C/min samples appear to have other small green structures in them besides the large shell (these small structures are very clear in Supp Fig 12).
4. It is not described how sizes of structures were measured, only that ImageJ was used for processing and analysis. This is particularly unclear for data in Fig 3a,b given the low resolution of structures in Fig 3c.
5. There is no indication of the number of independent fabrications of these structures (new batch of

protein for a new batch of particles) to demonstrate the reproducibility of the processes. The quantitative data on size appears to cover measurements of multiple particles taken from the same sample.

Minor:

1. The discussion section would be better termed a conclusion section. Only a statement of the proposed novelty and restatement of the main results are provided.
2. Supp Fig 1: What does "either" mean? It sounds like it is inconsistent and sometimes forms 1 phase and sometimes forms another. Or does it have to do with whether it is heating or cooling? This should be explained.
3. Fig 2c and Supp Fig4a are identical and the accompanying images are just slightly different magnifications of what looks like the same thing. What is the purpose of Supp Fig 4a? The same is true of Fig 2f,i and Supp Fig4b,c and accompanying images. Also, in Supp Fig 4 above each plot are titles with E and H in them that are not explained.
4. What is the inset of Fig 4biv?
5. There is no SDS-PAGE gel to demonstrate purity of protein polymers.

Reviewer #2 (Remarks to the Author):

The authors have described an elegant procedure to create microparticles with customizable internal architectures by combining thermally responsive IDPs within microdroplets. By balancing the phase behaviour of the constituents with respect to the proportion of the formulation, the size of the pores, surface textures and networks can be engineered. The interface of biomolecules and soft matter physics in this context is innovative in which new perspective can be drawn convincingly.

The study is comprehensively presented with systematic changes with high scientific relevance to the community. With this attractive approach, the demonstration of its potential is somewhat thin and can be reinforced to make the final point more compelling to the broader audience.

In this context, a standard experiment that takes advantage of the stimulus responsive behaviour in terms of size or internal architectures would be beneficial. It could be a release mechanism for self-healing for example. Or from a materials perspective, the particle rheological properties of some of these particles would also be very interesting. Either one of such demonstration (or others) would suffice to round up the story.

Other points:

1. On page 6, the authors stated that the average volume of the globules increases linearly with the concentration of the ELP in solution.... (Supporting figure 2). If the authors would really like to prove a linear relationship, I would very much a few more data points.
2. Although, in the methods, it is stated that the experimental group sizes are presented in each individual experiment. Most of these are missing in both the main figures and supporting figures (with the exception of S5 and S10). Sample sizes are particularly important in this manuscript as the particles are, in majority, counted manually with microscopy. The sample size should be also be large enough to be representative.

Overall, I have enjoyed reading the manuscript and would recommend publication after the above minor revision.

David Ng

Revisions to be made in response to the comments of Reviewer #1:

Summary comments:

This manuscript describes the fabrication of protein microparticles with complex internal structures by basic microfluidic processing techniques. Two polypeptides that exhibit temperature induced phase transitions are used and the differences in the phase transition temperatures of different protein variants, as well as protein concentration and heating rate, are used to produce several different microstructures. The ability to produce several different protein microstructures with the same class of molecules is unique, though the authors do not acknowledge prior reports of similarly structured protein microparticles made from different proteins and simple processes. Without demonstration of the utility of any of the reported structures, it is difficult to gauge the impact of these particular materials relative to more simple protein or polymer microparticles with no internal structure that are common in the literature, or the few reports of complex-structured protein microparticles. It may be possible to extend this phase transition methodology to other types of polymers and proteins that have temperature (or other) triggered transitions, thereby providing a more general framework for complex microstructure formation. Specific comments are listed below.

Response to summary comments:

We would like to thank the reviewer for their overall positive view of both the uniqueness of this system and its ability to provide an innovative framework within this field. We have addressed their specific critiques on a point-by-point basis below. Appropriate changes have also been made in accordance with the reviewer's request to the accompanying revised version of the manuscript.

Comment 1:

While the geometries are beautiful, it is not demonstrated or described what value these protein microparticles have. It is mentioned that there are "potential applications in drug delivery and tissue engineering" but there is no discussion of why these complex morphologies would be needed relative to simple spherical protein or biopolymer/carbohydrate hydrogel microparticles that are so prevalent in the literature, or practical demonstration of the utility of complex microstructure. Presumably, to be used in bio-applications the microparticles will need to be loaded with functional molecules or cells in a way that is useful relative to the local micro-geometry. Will these additives affect the ability to form the structures?

Response to comment 1:

To the reviewer's first point, we humbly submit that the vast field of creating complex microparticles using flow lithography and complex microfluidics would not exist if simple spherical microparticles could already solve all issues in biotechnology. Rather than developing a single morphology for a specific function, we, as the reviewer notes, focused this manuscript on the creation of a variety of complex structures using a single, novel protein system. Drug delivery and tissue engineering were specified as those are the most common biotechnology uses for the base elastin material. While we maintain that the novelty, and therefore focus, of the manuscript should be on the diversity of achievable architectures, in accordance with the requests of reviewer 1 and 2 we have included additional analysis on the mechanical properties of the

microparticles using AFM as well as a functional demonstration of a POP-ELP network as a drug eluting scaffold. To the reviewer's second point, utility of these systems will almost certainly come from their recombinant nature. Bioactive proteins and interaction domains can and have readily been included in other elastin-based systems by our group and several others with minimal, if any, impact on material properties.

Comment 2:

The first half of the following sentence (in the "Discussion"), from which the authors derive novelty, is not true. "All of these architectures are novel in protein- or polymer-based microparticles ..." The manuscript did not cite previous work making protein microparticles with complex morphologies under simple conditions with morphologies similar to those described here. A literature search finds relevant three papers from the last 7 years, including microclusters of protein spheres (Song. "Budding-like division of all-aqueous emulsion droplets modulated by networks of protein nanofibrils." Nature communications 9.1 (2018): 2110.), hollow shell and core shell protein microstructures (Park. "Thermally triggered self-assembly of folded proteins into vesicles." Journal of the American Chemical Society 136.52 (2014): 17906-17909.), and porous protein microparticles (Volodkin. "One-Step Formulation of Protein Microparticles with Tailored Properties: Hard Templating at Soft Conditions." Advanced Functional Materials 22.9 (2012): 1914-1922.) While the literature in this area is sparse and describes single or dual morphologies, and this manuscript is capable of making multiple morphologies with the same general system, previous work should be acknowledged and used to place the current work into context. Indeed, as noted in this manuscript, in the polymer field published complex microstructures seem to be the result of complex processes.

Response to comment 2:

The reviewer raises an interesting point and perhaps one we should have handled more carefully. We were aware of each of the publications listed in this notably sparse field, but did not include them in our discussion for different reasons. The Song and Volodkin papers use dextran/PEG and CaCO₃ respectively in the formation of their particles. Both manuscripts also create single architectures which are reminiscent of, but by no means analogous to the architecture we create. Park and Champion do form a type of protein-based vesicle, but these structures are only a couple of microns in diameter and form using a type of amphiphilic self-assembly. We, however, do not dispute the reviewer's reasons for wanting these works to be cited and have happily included citations of them in the manuscript. **The fact that we are capable of capturing all current morphologies and expanding to new unique structures using a single system is a notable feature of our approach.**

Comment 3:

It is very difficult to tell that the images and cartoons in Fig 3c match. The POP structures appear very fuzzy. There is no visually detectable difference between 20 and 6 degrees C/min heated samples. The smaller shell drawn in the cartoon and reported in Fig 3a,b is only visible in one particle for 1 degree C/min heating. The 0.5 degree C/min samples appear to have other small green structures in them besides the large shell (these small structures are very clear in Supp Fig 12).

Response to comment 3:

We apologize to the reviewer for any difficulty in discerning the details in the images. To help alleviate this concern we have added additional single plane confocal images of magnified particles to Sup. Fig. 15 that help to clarify the observed architectures. For the 1 C/min sample, the smaller shell(s) is actually present in 6 of the droplets in the original image (Response Fig. 1). We have further increased the contrast of this image in its original place in Figure 3 to aid in visualization. The 0.5 C/min sample does, as the reviewer notes, have other smaller green structures surrounding the hollow-shell. This observation was addressed in Sup. Fig 15, where they are even more clearly seen in the 3D confocal stack images. We believe these small structures arise for two reasons: (1) one is simply the minimal surface area available to interact with the templating ELP core. (2) The second reason is more nuanced.

The POPs phase separate into a two phase system in which a very small amounts of the polymer remains soluble above the T_t . Photo-crosslinking the dense phase of the proteins does not crosslink the dilute phase and, during additional heating and cooling cycles, they can be seen to aggregate into these small green structures. In any event, these structures readily wash away during extraction, and therefore have no impact on the final, desired product.

Response Figure 1: POP shells created from 1 C/min heating rate. Arrows point to smaller, secondary hollow shells.

Comment 4:

It is not described how sizes of structures were measured, only that ImageJ was used for processing and analysis. This is particularly unclear for data in Fig 3a,b given the low resolution of structures in Fig 3c.

Response to comment 4:

Whenever possible, the process for quantification was automated as much as possible to reduce experimental bias. We've provided two examples for the reviewer in Response Figure 2 that describe our use of standard utilities in ImageJ to isolate and measure size. Each separate imaging experiment required slight modifications, but these examples represent the general processing conditions used. We have also elaborated on the methods used in the methods section of the manuscript. The reviewer raises a good point specifically about the faster heating rates in Figure 3. While quantification using the denser images was possible manually, we instead opted to use confocal images (Sup. Fig. 15) for size analysis. We have noted this in the Figure caption for clarity.

a

Automated Analysis of ELP Globule Size

Split Channel -> Threshold -> Make Binary -> Analyze particles (Size: 1- infinity, circularity 0.8-1) -> Export

b

Manual Analysis of Emulsion to Shell Size Ratio

Response Figure 2: Analysis strategies used in imageJ for size analysis of particles. Examples include: (a) ELP globules size from Sup. Fig. 12 and (b) hollow shell ratio analysis from Fig. 3d-e.

Comment 5:

There is no indication of the number of independent fabrications of these structures (new batch of protein for a new batch of particles) to demonstrate the reproducibility of the processes. The quantitative data on size appears to cover measurements of multiple particles taken from the same sample.

Response to comment 5:

This is a fair comment. We can confirm directly to the reviewer and the journal that reproducibility was of the highest priority considering the uniqueness of the structures, and all were produced several times to ensure consistency. While the author is correct that several of the size data points were taken from a single experimental batch, the statement '*all experiments were repeated at least three times with similar results*' in the methods section was not made without reason. Where possible (e.g., ELP globule size in Sup Fig. 3, shrinking and swelling in Sup Fig. 7, etc.) experiments were averaged from several independent batches of proteins. Any omission in our original paper regarding replicability was simply an oversight on our part.

Minor comment 1:

The discussion section would be better termed a conclusion section. Only a statement of the proposed novelty and restatement of the main results are provided.

Response to minor comment 1:

We do not disagree with the reviewer's preferred terminology and have re-named that section 'conclusion'.

Minor comment 2:

Supp Fig 1: What does "either" mean? It sounds like it is inconsistent and sometimes forms 1 phase and sometimes forms another. Or does it have to do with whether it is heating or cooling? This should be explained.

Response to minor comment 2:

The reviewer's second position is correct here. 'Either' indicates that the protein is soluble if heating from a colder temperature but remains aggregated if cooling from a warmer temperature. This explanatory statement has been added to the figure caption.

Minor comment 3:

Fig 2c and Supp Fig4a are identical and the accompanying images are just slightly different magnifications of what looks like the same thing. What is the purpose of Supp Fig 4a? The same is true of Fig 2f,i and Supp Fig4b,c and accompanying images. Also, in Supp Fig 4 above each plot are titles with E and H in them that are not explained.

Response to minor comment 3:

Supplemental Figure 6 (previously 4) is, as the reviewer pointed out, largely a duplicate of Figure 2. This figure was included to demonstrate that isolated particles were not cherry picked for Figure 2, and that the architectures presented there are consistent across all particles during a heat-cool cycle. The phase diagrams were re-included simply for reference for each stage of a cycle. A statement to this effect has now been included in Sup. Fig. 6's caption.

The (E) and (H) above the diagrams were a different naming convention we originally used for the proteins. This inclusion was a mistake and we thank the reviewer for their keen eye in catching it.

Minor comment 4:

What is the inset of Fig 4biv?

Response to minor comment 4:

The description for this inset was mistakenly omitted and has now been added to the Figure 4 caption. While 4b_{iii-iv} predominantly exist to show that single hollow shells can be extracted, the inset further demonstrates that hollow shell networks, created using faster heating rates, can be extracted using the same method.

Minor comment 5:

There is no SDS-PAGE gel to demonstrate purity of protein polymers.

Response to minor comment 5:

We thank the reviewer for pointing out this oversight. We have added an SDS PAGE gel to the Supplementary Figures.

Revisions to be made in response to the comments of Reviewer #2:

Summary comments:

The authors have described an elegant procedure to create microparticles with customizable internal architectures by combining thermally responsive IDPs within microdroplets. By balancing the phase behaviour of the constituents with respect to the proportion of the formulation, the size of the pores, surface textures and networks can be engineered. The interface of biomolecules and soft matter physics in this context is innovative in which new perspective can be drawn convincingly. The study is comprehensively presented with systematic changes with high scientific relevance to the community. With this attractive approach, the demonstration of its potential is somewhat thin and can be reinforced to make the final point more compelling to the broader audience. In this context, a standard experiment that takes advantage of the stimulus responsive behaviour in terms of size or internal architectures would be beneficial. It could be a release mechanism for self-healing for example. Or from a materials perspective, the particle rheological properties of some of these particles would also be very interesting. Either one of such demonstration (or others) would suffice to round up the story. Overall, I have enjoyed reading the manuscript and would recommend publication after the above minor revision.

-David Ng

Response to summary comments:

We thank the reviewer for their favorable assessment of our manuscript. We have made a few additions to the manuscript to address his point on rounding out the story. Specifically, we have included an *in vivo* animal study that demonstrates that bulk POP-ELP mixtures can be used effectively as injectable drug eluting scaffolds. We also performed high frequency atomic force microscopy on our extracted microparticles to better probe their mechanical properties with some surprising and interesting results.

Comment 1:

On page 6, the authors stated that the average volume of the globules increases linearly with the concentration of the ELP in solution.... (Supporting figure 2). If the authors would really like to prove a linear relationship, I would very much a few more data points.

Response to comment 1:

This is a fair point. The acquisition of data points outside of the presented range is more difficult as lower ELP concentrations are too small for this type of image analysis and higher concentrations have T_t 's that are low enough to begin disrupting the clean two-stage formation (POP aggregation followed by ELP aggregation). To ensure scientific integrity, we have altered the language in the manuscript to remove assumptions of a linear relationship.

Comment 2:

Although, in the methods, it is stated that the experimental group sizes are presented in each individual experiment. Most of these are missing in both the main figures and supporting figures (with the exception of S5 and S10). Sample sizes are particularly important in this manuscript as the particles are, in majority, counted manually with microscopy. The sample size should be also be large enough to be representative.

Response to comment 2:

We thank the reviewer for pointing out this lapse. We have added sample size to each individual figure and expounded on the differences in the statistics and reproducibility section of the methods.

REVIEWERS' COMMENTS:

Reviewer #1 (Remarks to the Author):

The authors responded well to both reviewers comments. Inclusion of an in vivo protein release experiment and mechanical properties, which in the case of modulus are quite different from controls, lend more support to possible utility of the reported unique structures. Inclusion of the most related, though distinct, few references in this general area help better support the uniqueness of the materials. Clarifications in figures and methods improved the manuscript. I recommend it for publication.

Reviewer #2 (Remarks to the Author):

The authors have addressed the relevant concerns. The publication can be accepted for publication.